# HOW HARD IS LEARNING TO CUT?
# TRADE-OFFS AND SAMPLE COMPLEXITY

**Sammy Khalife**[*]
School of Operations Research and Information Engineering
Cornell Tech, Cornell University
khalife.sammy@cornell.edu

**Andrea Lodi**
Jacobs Technion-Cornell Institute
Cornell Tech and Technion - IIT
andrea.lodi@cornell.edu

## ABSTRACT

In the recent years, branch-and-cut algorithms have been the target of data-driven approaches designed to enhance the decision making in different phases of the algorithm such as branching, or the choice of cutting planes (cuts). In particular, for cutting plane selection two score functions have been proposed in the literature to evaluate the quality of a cut: branch-and-cut tree size and gap closed. In this paper, we present new sample complexity lower bounds, valid for both scores. We show that for a wide family of classes $\mathcal{F}$ that maps an instance to a cut, learning over an unknown distribution of the instances to minimize those scores requires at least (up to multiplicative constants) as many samples as learning from the same class function $\mathcal{F}$ any generic target function (using square loss). Our results also extend to the case of learning from a restricted set of cuts, namely those from the Simplex tableau. To the best of our knowledge, these constitute the first lower bounds for the learning-to-cut framework. We compare our bounds to known upper bounds for neural networks in the unsupervised setting and show they are nearly tight, suggesting that both scores (gap closed and tree size) are of comparable difficulty from a learning standpoint. Guided by this insight, we provide empirical evidence – by using a graph neural network cut selection evaluated on various integer programming problems – that gap closed is a practical and effective proxy for minimizing the tree size. Although the gap closed score has been extensively used in the integer programming literature, this is the first principled analysis discussing both scores simultaneously both theoretically and computationally.

## 1 INTRODUCTION

Branch-and-cut algorithms form the cornerstone of integer programming solvers. In recent years, machine learning has been playing a growing role in enhancing those solvers by enabling data-driven decision-making in various components of the algorithm. Recent attempts aim at augmenting those solvers, which often rely on handcrafted heuristics, by training models on data obtained from solved instances, to predict decisions that lead to faster convergence (which cutting plane – or cut, for short – to choose, or which variable to branch on). Specifically referring to cuts, there has been a growing body of work recently. Paulus et al. (2022) proposed a neural architecture that employs imitation learning to select cutting planes in mixed-integer linear programs (MILPs). By mimicking a lookahead expert that evaluates the potential impact of cuts on future bounds, their method aims to improve the efficiency of cut selection. In Huang et al. (2022), the authors trained a neural network to learn a scoring function evaluating the quality of candidate cuts based on instance-specific features. Tang et al. (2020) explored the use of deep reinforcement learning to adaptively select cutting planes in integer programming. By formulating cut selection as a Markov Decision Process, their method trains an agent to make the right cut selection among the Tableaux cuts. Subsequently, Ling et al. (2024) addressed the challenge of determining when to stop generating cuts, using reinforcement learning and different features of MILPs to make informed decisions. We refer the reader to the excellent survey Deza & Khalil (2023) for a more exhaustive list on previous contributions.

---

[*]Published while at Amazon, New York, NY, USA. This work was conducted entirely at Cornell Tech.

A fundamental question in any learning-based approach for generating cutting planes or making branching decisions during the solving process is how many training samples are needed to ensure good performance across an entire (and potentially unknown) distribution of problem instances. This issue – referred to as *sample complexity* – is critical, as it determines the scale of the learning task and directly impacts the feasibility of effectively training models. Understanding sample complexity helps address the inherent challenges of data-driven approaches by indicating how many instances must be solved to learn patterns that generalize reliably across a distribution. Establishing such bounds is therefore central to providing rigorous scientific foundations for data-driven techniques in integer programming. For instance, learning effective cut generation might require so many samples that the approach becomes impractical, or, in a more favorable case, only a moderate and affordable number of samples, scaling reasonably with the number of variables and constraints. Recent upper bounds have been obtained in the case of a fixed cut selection across a distribution of instances, Balcan et al. (2021), and more recently in the setup where neural networks map an instance to a cut, with a polynomial-logarithmic dependence on the size of the problems Cheng et al. (2024). Our investigation asks whether those bounds are tight for broader architectures and how the given ways of measuring the effectiveness of the cut affect sample complexity. If one metric were intrinsically harder to learn than another, this would shape both the design of learning algorithms and the choice of evaluation criteria. Our lower bounds provide clarity here, showing that – at least for unknown distributions – gap closed and tree size exhibit comparable sample complexity, thus preventing misleading conclusions about relative difficulty.

While these theoretical insights are important, they do not resolve on their own the practical challenges of implementing learning-based strategies. In particular, even if two performance measures are comparable from the standpoint of sample complexity, the computational effort required to optimize them may differ substantially. Our second contribution addresses this gap by showing that the gap closed score, despite being a proxy, aligns closely enough with tree size to serve as a practical surrogate. Taken together, these results provide a coherent picture: the first establishes that gap closed and tree size are theoretically equivalent in terms of learnability, while the second demonstrates that gap closed is a computationally tractable and empirically reliable alternative to tree size for guiding cut selection. In summary, the motivation for our work stems from the following two groups of observations, leading to two main results.

1. The existing studies applicable to sample complexity of learning-to-cut provide upper bounds for specific learning algorithms, formally referred to as *concept classes*. Those studies are applied to a special family of cutting planes, namely Chvátal-Gomory (CG) cuts Gomory (1958); Chvátal (1973). Specifically, in Balcan et al. (2021), the concept class is restricted to functions that return *constant* CG weights applied to any instance. In Cheng et al. (2024), the CG weights are generated by a neural network taking as input an integer linear program (ILP) instance.
   *Our contribution* is to provide the first quantitative lower bounds on sample complexity, and study lower bounds that are valid for a wide family of classes. Our lower bounds are discussed in Section 3 and anticipated in Table 1.

2. There are two main scores proposed in the literature to evaluate the quality of a cut. The first one is based on the relative size reduction (or increase) of the branch-and-cut (B&C) tree size. The second one is the relative improvement in the objective function of the relaxed problem (gap closed, where the gap for a MILP is the relative difference between the value of its linear programming, LP, relaxation and that of its optimal solution). The first score correlates well with the overall running time of the algorithm as it corresponds roughly to the number of LPs solved. However, it is easy to see that it is very expensive to train using the tree size because it requires to solve the problem to optimality to be evaluated. So, the second one could be considered as a proxy of the first, and the natural question we aim at discussing is how good the proxy is both in theory and in practice.[1]
   *Our contribution* is to empirically show the quality of the gap closed proxy and assess the ability of a graph neural network to learn both score functions in practice. Although the gap closed score has been extensively used in the integer programming literature, this is the first principled analysis discussing both scores at the same time both theoretically and computationally. The computational evaluation is conducted in Section 4.

---

[1]From the theory side, the upper bounds in Balcan et al. (2021); Cheng et al. (2024) are obtained for the branch-and-cut tree size score, although similar approach would yield the same upper bound for both scores.

Table 1: Illustration of sample complexity bounds in the case of ReLU neural networks with $W$ weights and $L$ layers, for IP instances with $n$ variables and $m$ constraints, verifying $M \geq \sum_{i=1}^{m} \sum_{j=1}^{n} |A_{ij}| + \sum_{i=1}^{m} |b_i|$. Here, $\overline{W} = W - w_1(n+1)m$ where $w_1$ is the number of neurons in the first hidden layer. The bounds in blue are our main theoretical contribution. The upper bounds were obtained in the unsupervised setup of Cheng et al. (2024), see Remark 2.1 for a discussion of the differences with our supervised formulation.

| Setting | B&C tree or gap closed scores | |
| --- | --- | --- |
| | Lower bound (supervised) | Upper bound (unsupervised) |
| all CG-cuts | $\Omega(\overline{W}L\log(\frac{\overline{W}}{L}))$ | $\mathcal{O}(LW\log(U+m) + W\log M)$ |
| tableau cuts | $\Omega(\overline{W}L\log(\frac{\overline{W}}{L}))$ | $\mathcal{O}(LW\log(U+t))$ |

Our first contribution sheds light on the learning difficulty of generating CG cuts from an instance when using two of the most common and theoretically grounded scores: gap closed and tree size. In particular, for classes such as neural networks, our bounds show that – absent further assumptions on the distributions – it is not theoretically harder to learn one score than the other, as our lower bounds nearly match known upper bounds. Our results can be put more broadly in the spectrum of *algorithm selection*, where selecting algorithms based on specific instances is allowed. For example, this is the case of Rice (1976); Gupta & Roughgarden (2016) where the sample complexity of learning mappings from instances to algorithms for particular problems is explored. Our approach is also related to recent work on algorithm design with predictions, see, e.g., Mitzenmacher & Vassilvitskii (2022) and the references therein.

Our second contribution highlights that, beyond this insight, the gap closed score remains a useful practical proxy for minimizing the more challenging tree size score over distributions of instances. For example, in the case of cuts generated by neural networks, our bounds indicate that there is no significant theoretical distinction between the two scores, and our numerical experiments further support that gap closed is a reasonable proxy across a wide range of instance families for cut selection.

The remainder of the paper is organized as follows. In Section 2, we properly define ILPs and its most successful solution method, i.e., branch and cut, as well as we give the basic definitions of learning theory. In Section 3, we discuss our main theoretical result on sample complexity lower bounds. An outline of the the proofs is given in Appendix A, and full proofs are deferred to Appendix B. In Section 4, we report on the computational investigation involving the two different score functions to evaluate cut quality. The full description and report of our experiments is included in Appendix C, due to space limitations. Finally, in Section 5, we draw some conclusions and outline open research questions.

## 2 PRELIMINARIES

In this section, we provide preliminaries for both ILP cutting plane methodology and learning theory.

### 2.1 BRANCH AND CUT AND CUTTING PLANES

We consider the ILP in the form

$$\max\{\mathbf{c}^\mathsf{T}\mathbf{x} : A\mathbf{x} \leq \mathbf{b}, \mathbf{x} \geq 0, \mathbf{x} \in \mathbb{Z}^n\}, \tag{1}$$

where $m, n \in \mathbb{N}_+$, and $A \in \mathbb{Q}^{m \times n}$, $\mathbf{b} \in \mathbb{Q}^m$, $\mathbf{c} \in \mathbb{R}^n$.[2]

The algorithms implemented in every (M)ILP solver are variations of a framework called *branch and cut*. In that algorithm, each iteration maintains: 1) a current best (integral) solution guess,[3] and 2) a list of polyhedra, each a subset of the original ILP relaxation. At each step, one polyhedron is selected and its continuous LP solution is computed. If the objective is worse than the current guess, the polyhedron is discarded. If the solution is integral, the guess is updated and the polyhedron is

---

[2]The ILP equation 1 is called MILP if a subset of the variables is allowed to take continuous values.
[3]Such guess would likely be $-\infty$ initially.

removed. Otherwise, the algorithm either adds *cutting planes* – valid inequalities that tighten the polyhedron – or *branches*. In branching, a variable $\mathbf{x}_i$ whose current value $\mathbf{x}_i^*$ is fractional is chosen, and the polyhedron is split using $\mathbf{x}_i \leq \lfloor \mathbf{x}_i^* \rfloor$ and $\mathbf{x}_i \geq \lfloor \mathbf{x}_i^* \rfloor + 1$. These two new polyhedra replace the original one. This process builds a branch-and-cut tree, with each node representing a polyhedron. The algorithm stops when the list is empty, returning the best guess as optimal. Often, a bound $B$ is set on the tree size; if exceeded, the algorithm terminates early and returns the current best guess.

There are many different strategies to generate cutting planes in branch-and-cut Conforti et al. (2014); Nemhauser & Wolsey (1988); Schrijver (1986). The oldest one is due to Gomory Gomory (1958) and later generalized by Chvátal Chvátal (1973), so the family of resulting cutting planes is called Chvátal-Gomory cuts. Namely, for any $\mathbf{x} \in \mathbb{Z}^n$ satisfying $A\mathbf{x} \leq \mathbf{b}$, then the inequality $\mathbf{u}A\mathbf{x} \leq \lfloor \mathbf{ub} \rfloor$ is valid for $S$ for all $\mathbf{u} \geq \mathbf{0}$ such that $\mathbf{u}A \in \mathbb{Z}^n$ and is called a CG cut. Gomory suggested to read $\mathbf{u}$ as the inverse of the basis of the tableau when the LP relaxation is solved by the Simplex method Gomory (1958). Chvátal generalized the procedure to any $\mathbf{u}$ Chvátal (1973).

Since the number of CG cuts that can derived at any iteration of the branch-and-cut algorithm is very large, any MILP solver implements its own cut selection strategy, i.e., decides which cuts are added to the current LP relaxation. The cut selection is performed by sophisticated, handcrafted heuristics and, as anticipated, the use of modern statistical learning to enhance these heuristics has been recently studied. The standard approach that has been used and that we inherit here is to decide the *single* next cut to be added within the CG family (or part of it). To do so, we need a score function that evaluates the quality of the cut, and two such functions have been investigated. Ideally, the branch-and-cut tree size *after* the addition of the cut is the right measure since most of the computing time is spent on solving the individual LPs in the nodes of the algorithm. However, this scoring function is very expensive to evaluate and, so far, has been used for theoretical purposes only. Instead, MILP technology generally measures the quality of a cut using the gap closed, i.e., the measure of the improvement of the LP relaxation after the addition of the cut. Of course, this is cheaper to evaluate (requires to solve one single LP per cut), but still too expensive in practice for performing cut selection, so the idea of *learning* such a score.[4]

It is interesting to note that, although the gap closed could be seen as a proxy of the branch-and-cut tree size, the two scores are hard to properly compare. More precisely, a cut could reduce significantly the tree size without even cutting off the optimal (fractional) solution of the LP relaxation, while a cut that does cut it off could have no effect long term, i.e., in reducing the tree size.

For example, consider the ILP $\{\max 5x_1 + 8x_2 \mid x_1 + x_2 \leq 6, 5x_1 + 9x_2 \leq 45, x_1, x_2 \geq 0, x_1, x_2 \in \mathbb{Z}\}$, whose fractional solution is $x^* = (\frac{9}{4}, \frac{15}{4})$. It can be shown that one of the CG cuts derived from the optimal tableau leads to the constraint $4x_1 + 7x_2 \leq 35$. Adding this constraint leads to a new fractional solution $(\frac{7}{3}, \frac{11}{3})$, located on the right (i.e., with greater $x$-coordinate) of the solution of the original formulation. Hence, supposing branching is performed first on $x_1$ then $x_2$, this leads to a larger branch-and-cut tree, with more LPs to be solved. However, this CG cut actually cuts off the fractional solution, hence improves the gap closed score.

## 2.2 LEARNING THEORY

We are interested in a statistical supervised learning problem of the following form, given a fixed parameterized function class defined by some $h$ with output space $\mathcal{O} = \mathbb{R}$:

$$\min_{f \in \mathcal{F}} \; \mathbb{E}_{(I,s) \sim \mathcal{D}}[(h(I, f(I)) - s)^2], \tag{2}$$

for an unknown distribution $\mathcal{D}$, given access to i.i.d. samples $(I_1, s_1), \ldots, (I_t, s_t)$ from $\mathcal{D}$. We restrict to learning problems of a function $f$ to minimize a given functional measuring the quality of a cutting plane in a branch-and-cut type of algorithm, where $s$ is a score of a "best cut" (according to this score), collected for the instance $I$. In this problem, one tries to learn the best decision $f \in \mathcal{F}$ for minimizing an expected error compared to the best encountered "cut score", with respect to an unknown distribution from which samples are drawn. In our branch-and-cut framework, we assume that we have access to an oracle returning the performance of the cutting plane after adding it to the

---

[4]It is worth mentioning that no solver adds one cutting plane at a time, but cuts are instead added in groups, called rounds. Analyzing such a procedure would be way harder, so literature studies – as well as our paper – concentrate on this simplified version.

ILP instance, that will be accounted for in the choice of the function $h$. We are interested in two performance scores: (i) the relative variation of the size of the branch-and-cut tree after adding the cut, and (ii) the gap closed score. Both will be formally defined in Section 2.1.

**Remark 2.1.** Formulation equation 2 entails a subtlety. The labels $s$ are drawn jointly with the instances from the distribution $\mathcal{D}$, rather than being a deterministic function of $I$. In the learning-to-cut context, the score would ideally correspond to the optimal cut for each instance, which suggests a deterministic map between instances and scores. We argue that this supervised setup remains appropriate for two reasons:

(i) In practice, B&C tree sizes exhibit randomness (due to imperfect tie-breaking, often referred as performance variability in the literature Lodi & Tramontani (2013)), so scores naturally follow a distribution conditioned on the instance, justifying the joint distribution over $(I, s)$.

(ii) Alternatively, one may consider a formulation without randomness in the labels, where the score is a deterministic but unknown function of the instance belonging to a sufficiently complex family. In this case, the argument based on fat-shattering dimension (Theorem 2.4) transfers directly: the distribution is on the instances only, and the complexity of the labeling family ensures that no learning algorithm can distinguish all possible labeling functions from limited samples — regardless of the fixed distribution on instances.

Conversely, if the labeling mechanism is simple (e.g., scores arise from a restricted family with low intrinsic complexity), the concept class $\mathcal{F}$ may be unnecessarily large and the lower bound would overestimate the true sample complexity. We also note that this formulation differs from the unsupervised setup of Cheng et al. (2024) where the learning objective is driven solely by the score function, and not the labels. The upper bounds in Table 1 were obtained in that unsupervised framework.

In this context, a *learning algorithm*[5] $L$ for $\mathcal{F}$ is a function taking as input a fixed (but arbitrary) amount of samples, and returning a function in $\mathcal{F}$

$$L : \bigcup_{m=1}^{\infty} (\mathcal{I} \times \mathbb{R})^m \to \mathcal{F}$$

Given $\epsilon \in (0, 1)$, $\delta \in (0, 1)$, the *sample complexity of learning* $m_0(\epsilon, \delta) \in \mathbb{N}$ of $L$ is the smallest integer (allowed to be $+\infty$) such that for any $m \geq m_0(\epsilon, \delta)$, for any probability distribution $\mathcal{D}$ on $\mathcal{I}$, the algorithm $L$ evaluated at "test time" on instance $I$ is is in average close to the solution on the entire distribution up to $\epsilon$:

$$\left| \mathbb{E}_{(I, s_I) \sim \mathcal{D}}[(h(I, L(I_1, \cdots, I_m)(I)) - s_I)^2] - \min_{f \in \mathcal{F}} \mathbb{E}_{(I, s_I) \sim \mathcal{D}}[(h(I, f(I)) - s_I)^2] \right| < \epsilon$$

with probability $1 - \delta$ over i.i.d samples $I_1, \cdots, I_m$ drawn following $\mathcal{D}$. The following two definitions are needed in order to formulate a fundamental result concerning the sample complexity of learning.

**Definition 2.2** (VC-dimension of a real output concept class). For any positive integer $t$, we say that a set $\{I_1, ..., I_t\} \subseteq \mathcal{I}$ is shattered by a concept class $\mathcal{E}$ defined on $\mathcal{I}$ taking $\{0, 1\}$-values if

$$2^t = |\{(f(I_1), \ldots, f(I_t)) : f \in \mathcal{E}\}|$$

The *VC dimension of* $\mathcal{E}$, denoted as $\mathrm{VCdim}(\mathcal{E}) \in \mathbb{N} \cup \{+\infty\}$, is the size of the largest set that can be shattered by $\mathcal{E}$.

If $\mathcal{F}$ is a non-empty collection of functions from an input space $\mathcal{I}$ to $\mathbb{R}$. Let $\mathrm{sgn}(\mathcal{F}) := \{\mathrm{sgn}(f) \in \mathcal{F}\}$ where $\mathrm{sgn}(x) = \mathbf{1}_{x>0}$. Then, $\mathrm{VCdim}(\mathcal{F})$ is by definition $\mathrm{VCdim}(\mathrm{sgn}(\mathcal{F}))$ where we adopt the standard defintion of $\mathrm{VCdim}$ for $\{0, 1\}$-function described above.

**Definition 2.3** (Fat-shattering dimension). Let $\gamma > 0$. With the same notations as Definition B.3, we say that the function class $\mathcal{F}$ fat-shatters $I_1, \cdots, I_t$ with precision $\gamma$ provided there exists $r \in \mathbb{R}^t$ such that for every labeling $(y_1, \cdots, y_t) \in \{-1, 1\}^t$, there exists $g \in \mathcal{F}$, such that $g(I_i) \geq r_i + \gamma$ if $y_i = -1$ and $g(I_i) \leq r_i - \gamma$ if $y_i = 1$. In such conditions, $r$ is called the witness of the shattering. The fat-shattering dimension of $\mathcal{F}$ with precision $\gamma$, noted $\mathrm{fat}_{\mathcal{F}}(\gamma)$ is the size of the largest that can be fat-shattered by $\mathcal{F}$.

---

[5]In unsupervised learning, the domain is typically formed by $\cup_{m=1}^{\infty} \mathcal{I}^m$.

In the case of binary functions, VC-dimension gives a direct way to bound *from above and below* learning sample complexity (Anthony & Bartlett, 2009, Theorem 5.4). For real output functions, the pseudo-dimension remain useful to find an upper bound on *uniform convergence* (UC). Typically, UC requires the absolute difference between the empirical mean and the full expectation to be bounded from above by $\epsilon$ for every $f \in \mathcal{F}$ and for every distribution. However, the sample complexity of learning can be smaller than that of UC. This leads to the sample complexity of UC to be an upper bound on the sample complexity of learning via Empirical Risk Minimization (ERM), which is itself greater than the sample complexity of learning in general, as there could be other algorithms performing better than ERM. In other words, uniform convergence guarantees that ERM will perform well, since the sample average closely matches the true expectation across all hypotheses. Good performance from ERM can still occur without full uniform convergence, and there may exist other learning algorithms that outperform ERM.

Therefore, lower bounds on Pseudo-dimension or VC-dimensions mainly apply to UC, and do not necessarily reflect the true sample complexity of learning. This surprising gap was first highlighted in Shalev-Shwartz et al. (2009) and further explored in Feldman (2016). As a consequence, to obtain lower bounds of learning sample complexity, one cannot *a priori* use standard traditional lower bounds of VC-dimension, and the analysis has to be performed carefully depending on the concept class considered. In this article, we will rely on the following result giving a general lower bound on the sample complexity of learning.

**Theorem 2.4.** (Anthony & Bartlett, 2009, Theorem 19.5) Let $\mathcal{F}$ be a class of functions from $X$ to $[0, 1]$. Then for any $0 < \epsilon < 1, 0 < \delta < 10^{-2}$, any learning algorithm $L$ for $\mathcal{F}$ has sample complexity $m_L(\epsilon, \delta)$ satisfying for every $0 < \alpha < \frac{1}{4}$,

$$m_L(\epsilon, \delta) \geq \frac{\text{fat}_{\mathcal{F}}(\frac{\epsilon}{\alpha}) - 1}{16\alpha}$$

Thus, any learning algorithm will have to use at least $\frac{\text{fat}_{\mathcal{F}}(\frac{\epsilon}{\alpha}) - 1}{16\alpha}$ samples to guarantee that the average solution at test time, independently of the distribution, will be at most at $\epsilon$ distance from the best solution of the function class, with probability $1 - \delta$. Note that the lower bound is rigorously valid only when $\delta < \frac{1}{100}$ (and the bound becomes independent of $\delta$ in that regime). We refer the reader to Remark 2.1 for a discussion on the conditions under which this lower bound is meaningful in the B&C context.

## 3 STATEMENT OF RESULTS

For any positive integer $d \in \mathbb{Z}_+$, $[d]$ refers to the set $\{1, 2, \ldots, d\}$. The sign function $\text{sgn} : \mathbb{R} \to \{0, 1\}$, is defined such that for any $x \in \mathbb{R}$, $\text{sgn}(x) = 0$ if $x < 0$, and 1 otherwise. This function is applied to each entry individually when applied to a vector. The elementwise floor function $\lfloor \cdot \rfloor$ is used to indicate the rounding down of each component of a vector to the nearest integer.

### 3.1 OVER THE POOL OF ALL CG-CUTS

We first present results in the case where the generation of CG-cuts is unrestricted, i.e., except the limitations brought by the cut generation process, the whole pool of CG-cuts is considered. We assume the following structure on the underlying concept class $\mathcal{F}$: each function of $\mathcal{F}$ incorporates an encoder function to transform each ILP to be processed further. For Neural networks, an example of such an encoder is the concatenation of all the instance's numerical data into a single vector. In the case of Graph Neural Networks (GNNs), one can choose a graph based representation (cf. for example Chen et al. (2024)). For ease of presentation, we will suppose that the stacking encoder is used (our results naturally extend to every encoder which is surjective onto the domain of each coordinate function of $\mathcal{F}$), and functions of $\mathcal{F}$ have domain $\mathbb{R}^{n \times m+m+n}$ and codomain $\mathbb{R}^m$ where $n$ is the number of variables of the ILP, and $m$ its number of constraints.

> **Assumption 1.** $\mathcal{F}$ is a non empty concept class closed under translation of the input, i.e., for every $\mu \in \mathbb{R}^{n \times m+m+n}$, $f \in \mathcal{F} \implies x \mapsto f(x + \mu) \in \mathcal{F}$, and under scaling of the output of every coordinate, i.e., for every real $\lambda$ and $i \in [m]$, and $f = (f_1, \cdots, f_m) \in \mathcal{F}$ implies that $(f_1, \cdots, \lambda f_i, \cdots, f_m) \in \mathcal{F}$. Note that is true for (graph) neural networks (for any activation function that is not identically zero).

**Assumption 2.** (Same shattering power by restriction to some row). Let $r = m \times n + m + n$. For every $i \in [m]$ representing the index of the associated CG-weight, $c \mapsto \mathrm{VCdim}(\mathcal{F}_i[n](c))$ is constant (cf. Definition B.1, here $\mathcal{F}_i$ refers to the concept class formed by the $i-$ coordinate of $f \in \mathcal{F}$). This is for example true for (graph) neural networks with any activation function[6]. In those conditions, we refer to this constant as $\mathrm{VCdim}(\mathcal{F}[n])$.

**Definition 3.1.** Let $s : \mathcal{I} \times [0,1]^m \to \mathbb{R}$ be a score function, mapping each pair formed by an ILP instance and a weight vector of a CG cut to a real value. Let $\mathcal{F}$ be a concept class following assumptions described above. Let $\sigma' : \mathbb{R}^m \to [0,1]^m$ be a *squeezing function* so that $\sigma' \circ f$ (where $f \in \mathcal{F}$) returns a vector in $[0,1]^m$ used as weights of the CG-cuts. We also suppose that $\sigma'$ is continuous at $\frac{1}{2}$ and verifies $\sigma'((-\infty, 0)) \subset [0, \frac{1}{2})$, $\sigma'([0, +\infty)) \subset [\frac{1}{2}, 1]$ and $(0, 1) \subset \sigma'(\mathbb{R})$. Let $\mathcal{F}_{\sigma'}$ be the concept class obtained. We define $\mathcal{F}_{s,\sigma'}$ as the final resulting concept class

$$\mathcal{F}_{s,\sigma'} := \{I \mapsto s(I, h(I)) : h \in \mathcal{F}_{\sigma'}\}$$

**Theorem 3.2.** Under those assumptions, for both gap-closed and branch-and-cut tree size scores, the sample complexity of learning CG-cuts via the class $\mathcal{F}_{s,\sigma'}$ verifies

$$m_L(\epsilon, \delta) = \Omega(\frac{\mathrm{VCdim}(\mathcal{F}[n])}{\epsilon})$$

According to the notion of learnability, Theorem 3.2 provides a lower bound on the minimum number of samples required to guarantee with probability $1 - \delta$ that for any distribution $\mathcal{D}$, the solution of *any learning algorithm* (in particular, this is true for the Empirical Risk Minimizer (ERM) algorithm) returns a solution whose predictions are at most $\epsilon$ far from the optimal neural network with high probability over the entire distribution.

**Corollary 3.3.** For any concept classes verifying Assumptions 1 and 2, $m_L(\epsilon, \delta)$ is bounded from below by the sample complexity of learning from $\mathcal{F}_n$ to a generic target function. In particular, with the same notation of Theorem 3.2, for every $\gamma > 0$, we have

$$m_L(\epsilon, \delta) = \Omega \left( \frac{\mathrm{fat}_{\mathcal{F}[n]}(\gamma)}{\epsilon} \right) = \Omega \left( \frac{\mathrm{VCdim}(\mathcal{F}[n])}{\epsilon} \right)$$

where similarly $\mathrm{fat}_{\mathcal{F}[n]}(\gamma) := \max_{i \in [m]} \mathrm{fat}_{\mathcal{F}_i[n]}(\gamma)$.

Corollary 3.3 applies in particular to neural networks (and to graph neural networks as well), up to adding an extra neuron on each layer.[7]

We now compare to the known upper bound in the case of neural networks (i.e., when $\mathcal{F}$ is composed of neural networks of a certain depth and width). The upper bound of the pseudo-dimension of this concept class given by (Cheng et al., 2024, Proposition 3.3) is $\mathcal{O}\left(LW\log(U + m) + W\log M\right)$ for ReLU neural networks and a squeezing function to constrain their outputs in $[0, 1]$, $M$ is an upperbound on the coefficients in $A$ and $b$, where $U$ is the *size* of the neural network, defined as $w_1 + \cdots + w_W$, and are also imposed the conditions that $\sum_{i=1}^m \sum_{j=1}^n |A_{ij}| \le a$ and $\sum_{i=1}^m |b_i| \le b$ for any $(A, \mathbf{b}, \mathbf{c}) \in \mathcal{I}$, and $M := 2(a + b + n)$.

Hence, ignoring logarithmic factors in $\frac{1}{\delta}$ and $\frac{1}{\epsilon}$, the best known upper bounds for $m(\epsilon, \delta)$ is given by $\mathcal{O}\left(\frac{1}{\epsilon^2}\left(LW\log(U + m) + W\log M\right)\right)$, for the BC tree size score. Since the result only use the invariance by the number of regions where the CG-cuts remain constants, their proof can adapted for the gap closed score, although we suspect that a better upper bound should be achievable in that case.

We now state our lower bound in the case of neural networks in the next proposition. Note that our lower bound does not use any amplitude on the input data of the problem.

**Proposition 3.4.** Suppose $\mathcal{F}$ is composed of ReLU neural networks with $\le L$ layers, and $\le W$ weights, with the concatenation encoder $I \in \mathcal{I} \mapsto (A, b, c) \in \mathbb{R}^{n \times m + m + n}$. There is a universal constant $C$ such that the following holds. Suppose $W > CL > C^2$ Consider both gap-closed and branch-and-cut tree size scores. Let $\overline{W} := W - w_1(n + 1)m$. Then, the sample complexity of learning CG-cuts via the class $\mathcal{F}_{s,\sigma'}$ verifies

$$m_L(\epsilon, \delta) \ge \frac{1}{\epsilon C} \overline{W} L \log \left( \frac{\overline{W}}{L} \right)$$

---

[6]This can be seen by adjusting the bias of the neurons in the first layer.

[7]There is no asymptotic difference between Pseudo-dimension and VC-dimension of real output neural networks, up to adding one layer or one neuron per layer.

A few comments are in order:

- The correction term of $w_1(n+1)m$, where $w_1$ is the number of neurons in the first layer, accounts for the restriction of the concept class to $n$ inputs, $\mathcal{F}[n]$. Our approach "ignores" $n \times m + m = (n+1)m$ inputs. This leads to an amount of $w_1(n+1)m$ weights that are being removed in the neural network.

- Recall that $W = \sum_{i=1}^{L} w_{i-1} w_i$ where $w_0 := n \times m + m + n$ is the input dimension, and the other $w_i$'s are the widths (number of neurons) of the Neural network considered on each layer. In particular $\overline{W} = W - w_1(n+1)m$ is always positive, and furthermore the ratio $\frac{\overline{W}}{W}$ is greater than $1 - \frac{w_1 w_0}{1+W} \geq 1 - \frac{W}{1+W}$.

- The lower bound supposes some structure on the layers and parameters given by $W > CL$. This loss of generality does not take place in our proof technique, but in the bit-extraction technique to give a lower bound the VCdim of the class of neural networks Bartlett et al. (2019). Therefore, in order to remove that assumption, one would have to either obtain a general lower requiring no particular structure, or adopt an entirely different approach, specific to shattering ILPs, that would not require a general VC dimension lower bound on neural networks.

Hence, supposing a regime where the number of weights in the neural network are large compared to the variables $n$ and number of constraints $m$, the gap of is of order $\frac{1}{\epsilon}$, between our lower bound and the best upper bound, ignoring logarithmic factors in $\frac{1}{\delta}$ and $\frac{1}{\epsilon}$. In a general learning framework, this gap is inevitable: see for instances discussions in (Anthony & Bartlett, 2009, Section 19.5). We suspect that this gap transfers for learning CG cuts, if no further assumption is made on the distribution of instances.

## 3.2 OVER THE POOL OF ALL CG-CUTS FROM THE TABLEAU

We now restrict to the pool of CG-cuts obtained from the tableau, so the concept class has to be changed slightly. We show that despite our restriction, the sample complexity is still driven by the VC-dimension of the underlying concept class. To make this formal, we suppose the following structure: each function of the concept class is decomposable as the composition of a function that takes as input an ILP instance $I \in \mathcal{I}$ and returns the $m$ CG-cuts from the tableau $(a_1, b_1), \cdots, (a_m, b_m)$. This can be perfomed using the the simplex algorithm. We also suppose that each function $g \in \mathcal{G}$ maps $(I, a_i, b_i)$ to a real value. The cut selected to be added to the instance is the one maximizing each of the $m$ scores, the concept class after selecting the maximum is $\tilde{G}$ (ties are broken by alphabetical order of the constraints).

**Definition 3.5.** Let $s : \mathcal{I} \times [0,1]^m \to \mathbb{R}$ be a score function, mapping each pair formed by an ILP instance and a weight vector of a CG cut to a real value. Let $\mathcal{G}$ be a concept class described above such that Assumptions 1 and 2 hold. We define $\mathcal{G}_s$ as the final resulting concept class $\mathcal{G}_s := \{I \mapsto s(I, g(I)) : g \in \tilde{\mathcal{G}}\}$.

**Theorem 3.6.** Under those conditions, for both gap-closed and branch-and-cut tree size scores, the sample complexity of learning CG-cuts via the class $\mathcal{G}_{s,\sigma'}$ verifies

$$m_L(\epsilon, \delta) = \Omega\left(\frac{\text{VCdim}(\mathcal{G}[n])}{\epsilon}\right)$$

**Proposition 3.7.** Suppose $\mathcal{G}$ is composed of neural networks with $\leq L$ layers, and $\leq W$ layers, with the concatenation encoder $I \in \mathcal{I} \to (A, b, c) \in \mathbb{R}^{n \times m + m + n}$. There is a universal constant $C$ such that the following holds. Suppose $W > CL > C^2$. Let $\overline{W} := W - w_1(n+1)(m+1)$. Then the sample complexity of learning CG-cuts via the class $\mathcal{F}_s$ from the optimal Tableau verifies

$$m_L(\epsilon, \delta) \geq \frac{1}{\epsilon C} \overline{W} L \log\left(\frac{\overline{W}}{L}\right)$$

In comparison with the upper bounds (Cheng et al., 2024, Corollary 2.8), ignoring logarithmic factors in $\frac{1}{\delta}$ and $\frac{1}{\epsilon}$, we have that $m(\delta, \epsilon) = \mathcal{O}\left(\frac{WL \log(Um)}{\epsilon^2}\right)$. where $U = w_1 + \cdots w_L$ is the total number of neurons. In the regime where the number of weights in the neural network are large compared to the variables $n$ and number of constraints $m$, which implies $\overline{W}$ to be of the order of $W$, our bound could be improved by integrating logarithmic factors in $m$ and $U$.

## 4 NUMERICAL EXPERIMENTS

Our experiments start by comparing two metrics: B&C tree-size reduction, the true but costly performance measure, and gap closed, a computationally cheaper yet noisier proxy. This raises the central question of whether training on gap closed can generalize to improvements in tree size. To investigate that, we model (ILP instance, cut) pairs with a GNN trained on CG cuts from the Simplex optimal tableau. The model is then used to predict cut quality, so as to guide cut selection. The entire methodology, including loss formulation and inference procedure, is provided in Appendix C.

### 4.1 EXPERIMENTAL SETUP

**Modeling ILP as GNN.** Each ILP instance augmented by a cut gets encoded by $E$ unambiguously as a weighted graph $G$ with a three dimensional feature vector on its vertices as follows: (i) The vertices of $G$ are split between the variables and constraint vertices. Each variable gets associated to a vertex, and each constraint as well, leading to a bipartite graph with $n + m$ vertices. Furthermore, each variable vertex receives a three-dimensional feature vector corresponding to the objective vector entry, plus the coefficient of the cut for that variable, as well as the right-hand side (same for all variables). The other vertices corresponding to constraints get the vector $(1, 1, 1)$ as feature (for dimensional homogeneity purposes). (ii) One edge is created between each variable vertex $i$ and constraint vertex $j$ provided the variable $i$ appears in constraint $j$. The associated edge has weight $a_{ij}$; the number of edges in the graph depends on the sparsity of $A$.

**Data.** We consider the very well-known Set Cover, Uncapacitated Facility Location, Knapsack, and Vertex Cover problems with their natural ILP formulations. The $1,000$ set cover instances have 50 subsets and 30 base elements. The $1,000$ uncapacitated facility location instances have 10 facilities and 10 clients. The knapsack instances have 2 knapsacks and 16 items. The vertex cover instances have 20 vertices and 50 edges. The details for randomly generating the instances are detailed in the supplementary material.

**Training.** The experiments were conducted on a Linux machine with a 24-core Intel Xeon Gold 6126 CPU, with 745Gb of RAM, and an NVIDIA Tesla V100-PCIE with 32GB of VRAM. We used Gurobi 12.0.1 Gur to solve the ILPs, with default cuts, heuristics, and presolve settings turned off. The GNNs were implemented using PyTorch 2.6.0 and Pytorch Geometric 2.6.1. The details of the implementation are detailed in the supplementary material.

### 4.2 EMPIRICAL RESULTS

The GNN is trained using the B&C tree size vs. gap closed as a proxy. We report the average tree size after adding the chosen tableau cut. A subset of the results is reported in Table 2, and full results are in Table 3 in Appendix C. For all benchmarks except the GNN one, only the branch-and-cut tree size is reported, as it is the final quantity of interest in this experiment. The results refer to $250$ test instances for each problem class. The table compares six strategies: the perfect predictor (Optimal) always using the CG tableau cut that results in the smallest B&C tree size, a classical heuristic that selects a cut according to its parallelism with respect to the objective function (Parallelism, see, e.g., Lodi (2009); Deza & Khalil (2023)), the cut efficacy (Efficacy), a mix of the cut efficacy and parallelism (Mix), a uniform random selection (Random), and the GNN using either the B&C tree size or the gap closed in training (GNN). The results show that the GNNs are able to learn and provide a solid improvement (facility location, knapsack) or stay on pair (set cover, vertex cover) with respect to a state-of-the-art cut selection heuristic (cf. full table in Appendix C). The GNN trained by the gap closed score function provides a good proxy, though there is room for improvement for both GNNs with respect to the perfect predictor.

## 5 DISCUSSION AND OPEN PROBLEMS

In this paper, we have presented the first sample complexity lower bounds on the learning-to-cut task and we have empirically analyzed the relationship between two score functions used to assess the quality of a cut. In the sample complexity bounds, no analysis was conducted on the cut candidates

Table 2: A subset of the results: average tree size on 250 test instances of the GNN trained using either the B&C tree size or gap closed as a proxy vs. four simple benchmarks.

(a) Set cover

| Setting | B&C tree | gap closed |
|---|---|---|
| Optimal | – | 4.95 |
| Parallelism | – | 8.29 |
| Efficacy | – | 9.90 |
| Mix | – | 9.30 |
| Random | – | 9.71 |
| GNN | 8.27 | 8.65 |

(b) Facility location

| Setting | B&C tree | gap closed |
|---|---|---|
| Optimal | – | 86.31 |
| Parallelism | – | 144.09 |
| Efficacy | – | 123.63 |
| Mix | – | 133.72 |
| Random | – | 152.46 |
| GNN | 128.85 | 134.61 |

that actually close the gap, i.e., cut off the fractional solution. This could give additional information to give better sample complexity bounds in the case of gap closed. Therefore, we conjecture that it is possible to obtain a better upper bound of the sample complexity for the gap closed score because, implicitly, a restricted number of cuts (only those cutting off the fractional solution) are required.

**On the setup and interpretation of the bounds.** As discussed in Remark 2.1, the upper bounds in Table 1 were derived in an unsupervised formulation Cheng et al. (2024), while our lower bounds apply to the supervised setup of Equation 2. The lower bounds are meaningful when the labeling mechanism exhibits sufficient complexity — either because scores arise from a distribution (reflecting solver nondeterminism), or because the labeling functions belong to a rich enough family. In entirely deterministic and structurally simple settings, the effective sample complexity could be lower than our bounds indicate.

## 6 ACKNOWLEDGMENTS

Both authors gratefully acknowledges support from the Office of Naval Research (ONR) grant N00014-24-1-2645.

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

## A  OVERVIEW OF THE PROOFS

We now give an overview of the proof of Theorem 3.2, and expand in Appendix B. A very similar approach will be used for Theorem 3.7. Our proof builds a collection of instances that can be shattered by the concept class that integrates the cut generating process as well as the scoring mechanism. We carefully handcraft instances making use of the assumptions made on the cut-generation mechanism, by integrating an uninformative constraint, used to be shattered by the underlying concept class $\mathcal{F}$.

The first observations and steps are:

- We start with a collection of vectors that are shattered by $\mathcal{F}$ (as many as the VC-dimension of $\mathcal{F}$). Since the concept class $\mathcal{F}$ is closed under scaling and translation, the entries of those vectors (redundant constraint vectors) can be considered to be non-negative. This allows us to ensure redundancy of the constraint.

- The VC-dimension coordinate invariance (Assumption 2) allows us to use the redundant constraint without loss of shattering power. Note that our approach could still work without that assumption, but this allows to make the original lower-bound statements as crisp as possible, in particular for the special case of neural networks, where upper bounds are known.

- We then *analyze* the space of weights of the corresponding CG cuts and the impact of the CG cuts on the increase/decrease of objective after adding one of those CG cuts. We focus on two particular regions that are sufficient to fat-shatter our instances. These regions are given by

$$\frac{1}{2} \leq u_2 \leq 1 - \frac{5}{36}\left(\frac{5}{2} + 2\gamma\right) \qquad \text{and} \qquad \frac{3 - (4 - \frac{5}{4} - \gamma)}{\frac{5}{2} + 2\gamma} \leq u_3 < \frac{1}{2}$$

$$\frac{1}{2} \leq u_2 \leq (1 - \frac{5}{16}) - \frac{\gamma}{4} \qquad \text{and} \qquad \frac{3 - (4 - \frac{5}{4} - \gamma)}{\frac{5}{2} + 2\gamma} \leq u_3 < \frac{1}{2}$$

where $\gamma$ is a *fixed* positive (but arbitrary) real between $0$ and $\frac{1}{2}$. Both regions of those CG weights lead to objective values that are at least $\Omega(\gamma)$ apart.

The key reason this approach succeeds in achieving the final goal is that the constraint vectors are first shattered by $\mathcal{F}$, yielding positive or negative outcomes for the redundancy constraints. In turn, this produces CG weights that lie in the distinct regions described above.

## B  PROOFS OF MAIN RESULTS

**Definition B.1** (Restriction of a concept class). Let $\mathcal{F}$ be a concept class (i.e., set of functions) from $\mathbb{R}^d \to \mathbb{R}$. For any $i \in [d]$ and $c \in \mathbb{R}^{d-i}$, we refer to $\mathcal{F}_i(c)$ as a shorthand for

$$\mathcal{F}[i](c) := \{x \mapsto f(x_1, \cdots, x_i, c) : f \in \mathcal{F}\}$$

*Proof of Theorem 3.2.* Theorem 2.4 guarantees that

$$m_L(\epsilon, \delta) \geq \frac{\text{fat}_{\mathcal{F}_{s,\sigma'}}(\frac{\epsilon}{\alpha}) - 1}{16\alpha}$$

holds for any $0 < \epsilon < 1$, $0 < \delta < 10^{-2}$ and $0 < \alpha < \frac{1}{4}$. Since $\mathcal{F}$ verifies Assumptions 1 and 2, we first use Lemma B.2 and select $\alpha = 4\epsilon$ with $\epsilon < \frac{1}{16}$ so that $\frac{\epsilon}{\alpha} \in (0, \frac{1}{2})$ to get

$$m_L(\epsilon, \delta) \geq \frac{\text{VCdim}(\mathcal{F}[n]) - 1}{64\epsilon} \geq \frac{\text{VCdim}(\mathcal{F}[n])}{128\epsilon}$$

where the last inequality holds provided $\text{VCdim}(\mathcal{F}[n]) \geq 2$. $\qquad \square$

**Lemma B.2** (Transfer Lemma). With the same notations and assumptions made on the concept class described in Subsection 3.1 and verifying Assumptions 1 and 2, then for every $\gamma \in (0, \frac{1}{2})$

$$\text{fat}_{\mathcal{F}_{s,\sigma'}}(\gamma) \geq \text{VCdim}(\mathcal{F}[n])$$

*Proof.* Let $0 < \gamma < \frac{1}{2}$ and let $r := \text{VCdim}(\mathcal{F}[n]) = \max_{i \in [m]} \text{VCdim}(\mathcal{F}_i[n])$. Without loss of generality, we will suppose that a coordinate maximizing the VC-dimension is the last one ($i = m$). Therefore, for every labeling $(y_1, \cdots, y_r) \in \{-1, 1\}^r$, there exists $g \in \mathcal{F}$ such that $(g(a_i))_m \geq 0$ if $y_i = 1$ and $(g(a_i))_m < 0$ if $y_i = -1$.

Assumption 1 guarantees that we can consider that the vectors $a_1, \cdots, a_r$ **do intersect the positive orthant** because the concept class is closed under translation of the input. This guarantees that we can restrict to a list of instances whose admissible region is in the positive orthant.

We construct $r$ instances, described by linear equalities, to be fat-shattered by $\mathcal{F}_{s,\sigma'}$ with margin $\gamma$ as follows:

$$P_i := \{x \in \mathbb{R}^2 : a_i^t \mathbf{x} \le 0,\ 2x_1 \le 4,\ 2x_2 \le \frac{5}{2} + 2\gamma,\ \mathbf{x} \ge 0\},$$

$$I_i := \max\{x_1 + x_2 : \mathbf{x} \in P_i,\ \mathbf{x} \in \mathbb{Z}^2\}.$$

Those instances can be lifted to $n$ variables and $m$ constraints, simply by adding redundant constraints and keeping the same objective. We retain the constraint under the form $2x_1 \le 4$ rather than simplifying it to $x_1 \le 2$, since this representation is more convenient for our choice of regions.

First, since the $a_i$'s are intersecting the positive orthant, the first constraint is redundant, and we will use the vectors $a_i$ to shatter the instances. For each instance, the objective of the relaxed problem at the optimum is $2 + \frac{5}{4} + \gamma$, and one solution is given by $x_1^* = 2$ and $x_2^* = \frac{5}{4} + \gamma$.

In the following, we suppose that $0 \le u_1 < \frac{1}{2}$ to eliminate the impact of the first constraint on the CG cut.

Consider the two regions in the $u_2, u_3$ space associated with the second and third constraint, giving rise to the CG cuts

- corresponding to the weights $\frac{1}{2} \le u_2 \le 1 - \frac{5}{36}\left(\frac{5}{2} + 2\gamma\right)$ and $\frac{1}{2} \le u_3 < \frac{20}{36}$. For each instance, this yields the inequality: $\lfloor 2u_2 \rfloor x_1 + \lfloor 2u_3 \rfloor x_2 \le \lfloor 4u_2 + u_3(\frac{5}{2} + 2\gamma) \rfloor \iff x_1 + x_2 \le 3$ since $\gamma < \frac{1}{2}$.

  Then, the two new vertices of the feasible region are $(2, 1)$ and $(2 - \gamma, 1 + \gamma)$, and for both of them the objective value is 3, so the amount of gap closed is $\frac{1}{4} + \gamma$ (the improvement ratio is $\frac{\frac{1}{4} + \gamma}{2 + \frac{5}{4} + \gamma} \ge \frac{\gamma}{5}$ since $0 < \gamma < 1$, i.e., here, the cut actually gives the integral solution).

- For any $\frac{1}{2} \le u_2 \le (1 - \frac{5}{16}) - \frac{\gamma}{4}$ and $0 \le \frac{3 - (4 - \frac{5}{4} - \gamma)}{\frac{5}{2} + 2\gamma} \le u_3 < \frac{1}{2}$, the CG-cut associated with $(u_1, u_2, u_3)$ yields the inequality $x_1 \le 3$: this cut is redundant, the solution is the same as before and the gap closed is 0.

Hence, we have two CG-cuts that yield for each instance two gap closed scores that are at least $\Omega(\gamma)$ away from each other.

**In the case of B&C-tree size score:** The same CG cuts can also be used for the B&C tree size score: on the one hand, it is clear that the branch-and-cut tree size after adding the first CG-cut is one (solving the LP only once gives an optimal solution that is integral). On the other hand, adding the redundant cut associated with the second cut at the root gives a branch-and-cut tree size of at least 3 nodes since one needs to branch at least once on a variable to obtain the integral solution. Therefore, we have two CG cuts that will yield two scores that are at distance 1 for any of the $n$ instances.

For any function $\tilde{g}$ in $\mathcal{F}$, $\tilde{g} : \mathbb{R}^8 \to \mathbb{R}^3$, we refer to $\tilde{g}$ as $\begin{pmatrix} A \\ b \\ c \end{pmatrix} \mapsto \begin{pmatrix} g_1(A_1, \cdots) \\ g_2(A_1, \cdots) \\ g_3(A_1, \cdots) \end{pmatrix}$, where $A_1$ is the first row of $A$. Since the vectors $a_i$ are shattered by $\mathcal{F}$, for every $y \in \{-1, 1\}^n$ and for every $i \in [n]$, there exists $\tilde{g} \in \mathcal{F}$ such that $\tilde{g}(P_i) = \begin{pmatrix} g_1(a_i, \cdots) \\ g_2(a_i, \cdots) \\ g_3(a_i, \cdots) \end{pmatrix} = \begin{pmatrix} q_i \\ r_i \\ \eta_i \end{pmatrix}$, where $\eta_i \ge 0$ if $y_i = 1$ and $\eta_i < 0$ if $y_i = -1$.

Above, we implicitly use Assumption 2 by supposing that the VC dimension of all the coordinates of the functions in $\mathcal{F}$, when restricted to the first $n$ entries, is the same. Remind that we now need to apply on top of $\tilde{g}$ the squeezing function $\sigma'$ to each coordinate. Using again the Assumption 1, we rescale the first and second component to 0 and add the appropriate bias to $g_1$ and $g_2$ so that the following conditions are satisfied (the following intervals correspond to the CG weights computed previously):

- Condition on $u_1 = \sigma'(q_i)$: for every $i \in [n]$, $0 \le \sigma(q_i) < \frac{1}{2}$ and $0 \le \sigma(q_i') < \frac{1}{2}$. This can be achieved by multiplying $g_1$ (hence $q_i$) by 0 and adding, for example, the bias $\sigma^{-1}(\frac{1}{4})$.

- Conditions on $u_2 = \sigma'(r_i)$:

    - If $y_i = 1$: $\sigma'(r_i) \in [\frac{1}{2}, 1 - \frac{5}{36}(\frac{5}{2} + 2\gamma)]$.
    - If $y_i = -1$: $\sigma'(r_i) \in [\frac{1}{2}, 1 - \frac{5}{16} - \frac{\gamma}{4}]$.

    Both can be achieved by multiplying the shattering function $g_2$ (hence $r_i$) by 0 and adding the bias $\sigma^{-1}(x_\gamma)$, where $x_\gamma := \min(1 - \frac{5}{36}\left(\frac{5}{2} + 2\gamma\right), 1 - \frac{5}{16} - \frac{\gamma}{4})$.

- Conditions on $u_3 = \sigma'(\eta_i)$: Let $\mu := \frac{3 - (4 - \frac{5}{4} - \gamma)}{\frac{5}{2} + 2\gamma}$. We add the bias $\mu$ to $g_3$ so that $\forall i \in [n]$:

    - If $y_i = 1$: $\quad \frac{1}{2} \le u_3 < \frac{20}{36}$.
    - If $y_i = -1$: $\quad 0 \le \frac{3 - (4 - \frac{5}{4} - \gamma)}{\frac{5}{2} + 2\gamma} \le u_3 < \frac{1}{2}$.

    With $u_1$ and $u_2$ verifying the above conditions, $u_3$ is the weight deciding which cut is being selected. Since $(0, 1) \subset \sigma'(\mathbb{R})$, $\sigma'((-\infty, 0)) \subset [0, \frac{1}{2})$, $\sigma'([0, +\infty)) \subset [\frac{1}{2}, 1]$ (cf. Definition 3.1), we only need to verify for some appropriate positive reals $\delta_1$ and $\delta_2$:

    - If $y_i = 1$: $\quad \eta_i \in [0, \delta_1]$ such that $\sigma'([0, \delta_1]) \subset [\frac{1}{2}, \frac{20}{36})$.
    - If $y_i = -1$: $\quad \eta_i \in [-\delta_2, 0)$ such that $\sigma'([-\delta_2, 0)) \subset [\frac{3 - (4 - \frac{5}{4} - \gamma)}{\frac{5}{2} + 2\gamma}, \frac{1}{2})$.

Such $\delta_1$ and $\delta_2$ exist by *continuity* of $\sigma'$ in $\frac{1}{2}$. Using closeness of $\mathcal{F}$ under scaling, we can multiply the function $g_3$ by the appropriate scalar to ensure those conditions.

In summary, when $y_i = 1$, the weights obtained after applying the squeezing function generate a first CG-cut $\mathbf{u}$ whose coordinates are in the first region, and when $y_i = -1$, the weights generating the CG-cut $\mathbf{u}'$ are in the second region.

In the first part of the proof, we have shown that the two corresponding regions in the CG cut weight space lead to two scores (for both the gap closed and tree size) that are $\Omega(\gamma)$ apart: therefore, the instances $P_1, \cdots, P_n$ with $n = \text{VCdim}(\mathcal{F})$ are $\gamma$-fat shattered (with a witness that depends on the score considered), so $\text{fat}_{\mathcal{F}_{s,\sigma'}}(\gamma) \ge \text{VCdim}(\mathcal{F}[n]) = \max_{i \in [m]} \text{VCdim}(\mathcal{F}_i[n])$. $\qquad\square$

**Definition B.3** (Pseudo-dimension). Let $\mathcal{F}$ be a non-empty collection of functions from an input space $\mathcal{I}$ to $\mathbb{R}$. For any positive integer $t$, we say that a set $\{I_1, ..., I_t\} \subseteq \mathcal{I}$ is pseudo-shattered by $\mathcal{F}$ if there exist real numbers $s_1, \ldots, s_t$ such that

$$2^t = |\{(\text{sgn}(f(I_1) - s_1), \ldots, \text{sgn}(f(I_t) - s_t)) : f \in \mathcal{F}\}|.$$

The *pseudo-dimension of $\mathcal{F}$*, denoted as $\text{Pdim}(\mathcal{F}) \in \mathbb{N} \cup \{+\infty\}$, is the size of the largest set that can be pseudo-shattered by $\mathcal{F}$.

*Proof of Corollary 3.3.* We use here the following fact (see Definition above) that for any concept class $\mathcal{F}$ with real outputs, $\text{Pdim}(\mathcal{F}) \ge \text{fat}_F(\gamma)$ for all $\gamma > 0$, see for example (Anthony & Bartlett, 2009, Theorem 11.13).

Also, by assumption each $\mathcal{F}_i[n]$ is closed under translation of the input so if a set of vectors $x_1, \cdots, x_n$ are pseudo-shattered, there are also shattered, leading to

$$\text{VCdim}(\mathcal{F}_i[n]) \ge \text{Pdim}(\mathcal{F}_i[n]) \ge \text{fat}_{\mathcal{F}_i[n]}(\gamma)$$

proving the claim. $\qquad\square$

*Proof of Proposition 3.4.* This is a direct application of Theorem 3.2 combined with state-of-the art VC-dimension lower bound for ReLU neural networks (Bartlett et al., 2019, Theorem 3). $\qquad\square$

*Proof of Theorem 3.7.* We perform a similar reasoning as in the proof of Theorem 2.4 with the following ingredients and similar notations:

- we first invoke Lemma B.4, that can be lifted up to more variables if needed. We create a collection of instances by adding to instance $i$ composed of the same instance of Lemma B.4, the redundant constraint $a_i^t \mathbf{x} \le 0$, where $a_i$ is in the nonnegative orthant.

- The underlying concept class maps the vector to one CG cut or the other (CG1 or CG2 in the proof of Lemma B.4), depending on the label for that instance.

This allows us to shatter a collection of instances by choosing one or the other CG cut, given the arbitrary $\{0, 1\}$ labels. □

**Lemma B.4.** There exists a two-variable ILP instance with two constraints such that the two cuts from the tableau have both scores (tree size and gap-closed) apart from a constant, positive distance.

*Proof.* Consider the following instance (same as in Section 2.1):

$$
\begin{aligned}
\max \quad & 5x_1 + 8x_2 \\
\text{subject to} \quad & x_1 + x_2 \le 6 \\
& 5x_1 + 9x_2 \le 45 \\
& x_1, x_2 \ge 0 \\
& x_1, x_2 \in \mathbb{Z}
\end{aligned}
$$

and the linear program obtained by relaxation of the above, where the last constraint is replaced by $x_1, x_2 \in \mathbb{R}^2$. Our goal is to show that two CG cuts derived from the tableau yields two improvements over the initial objective that are far apart by a constant value.

Ater two iterations of the simplex method, the optimal solution is given by $(\frac{7}{4}, \frac{17}{4})$, with an optimal cost of $42.75$. The final optimal tableau of the simplex method (in the original space of two variables) is given by $\begin{pmatrix} 4 & 7 & 35 \\ 2 & 3 & 15 \end{pmatrix}$, so that the two CG cuts are:

1. **(CG1):** $2x_1 + 3x_2 \le 15$. We list all the vertices obtained by intersecting with the two constraints. The first one obtained with the constraint $x_1 + x_2 = 6$, gives the vertex $(3, 3)$. The objective value at that point is $15 + 24 = 39$. The other vertex is $(0, 5)$ with an objective value of $40$, and a change of $1.25$ in objective (corresponding to the absolute gap closed). Since the other vertices $(0, 0)$, $(6, 0)$ have an objective of $0$ and $30$ respectively, hence $(0, 5)$ is the actual optimal integral solution (it has integral entries). Furthermore, the tree size after adding the cut is $1$, whereas is it at least $2$ without the cut.

2. **(CG2):** $4x_1 + 7x_2 \le 35$. Similarly, the two new vertices with greatest objective are $(\frac{9}{4}, \frac{15}{4})$, and $(0, 5)$, with objective value $41.25$ and $40$ respectively. The gap closed is $1.5$ in absolute value, and the tree size is at least $2$ after adding the cut (one needs to do an additional round of branching because the solution is fractional).

Therefore, the two CG cuts have a tree size and gap closed scores and gap close that are distant by a positive constant. □

*Proof of Proposition 3.7.* This is a direct application of Theorem 3.7 combined with state-of-the art VC-dimension lower bound for ReLU neural networks (Bartlett et al., 2019, Theorem 3). □

## C  NUMERICAL EXPERIMENTS

To start the sample complexity analysis computationally, we wish to investigate how both scores in the literature relate empirically, based on the premise that (i) the reduction in the B&C tree size is ultimately the score of interest but is costly to obtain and learn, and (ii) the gap closed is easier to compute but less reliable as a training signal for the end-task of minimizing the B&C tree size.

A potential trade-off emerges: cuts that close large gaps may not always reduce tree size due to some situations where both are incomparable (see end of Section 2.1 ), or from branching decisions that change the impact of one cut overall. This setup mirrors classic proxy optimization challenges in machine learning, where we want to learn for a costly target (tree size), but we use a cheaper, noisier proxy (gap closed), hoping for performance generalization to the target.

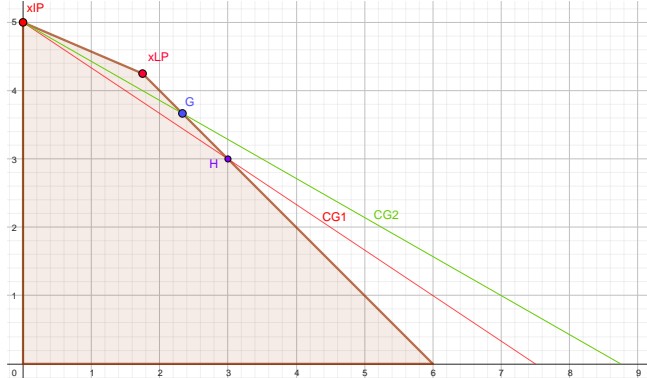

Figure 1: Example of 2D instance used to construct the lower-bound and prove Lemma B.4. The optimal fractional solution is $x_{\text{LP}} = (\frac{7}{4}, \frac{17}{4})$ with objective value $42.75$. Both cuts CG1 (red) and CG2 (green) are two CG cuts derived from the Optimal Tableau. The red cut gives directly the integral optimum $x_{\text{IP}} = (0, 5)$ (objective value of $40$), after solving one LP (the vertex $H = (3, 3)$ has an objective value of $39$). The green CG cut gives a optimal fractional solution $G = (\frac{9}{4}, \frac{15}{4})$ with objective value $41.25$, and at least one additional LP has to be solved to reach the integral solution. Therefore, both cuts lead to distinct (constant) improvements for both the tree size and the gap closed score. With respect to our original proofs in the tableau case, the vectors of the redundant constraints are mapped to one CG cut vector or the other, to $\gamma$-shatter the instances according to the score considered.

Our computational methodology is based on the two key building blocks: (1) We represent each pair (ILP instance, cut) as a graph, i.e., we encode variables and constraints by a GNN, with proper edges and features. GNNs naturally encode ILP instances well because the solution of an ILP does not depend on the order of the rows, which is captured by the isomorphism invariance of the associated representation. (2) At training time, we generate all the CG cuts from the optimal Simplex tableau with corresponding scores, for any considered ILP instance. The GNN is trained to match the scores returned for each CG cut via a cross-entropy loss.

Having collected the (up to) $m$ CG cuts from the optimal Simplex tableau, and their corresponding scores $s_1, \ldots, s_m$ (either gap closed or B&C tree size reduction), for each of the $t$ instances, we approximate a solution of the problem

$$\min_W \frac{1}{t} \sum_{i=1}^{t} \ell\big((H_W(E(I_i, o_i)))_{j \in [m]}, (s_j)_{j \in [m]}\big), \tag{3}$$

where $E$ is the instance and cut encoder (described in the next subsection), $H_W$ is a GNN parametrized by the weights $W$, which takes as input a graph and vectors $o_i$ of size $m+1$ representing the collected cut from the tableau (left-hand side and right-hand side), and $\ell$ is the cross entropy loss $\ell : \mathbb{R}^m \times \mathbb{R}^m \to \mathbb{R}, (x, y) \mapsto \ell(x, y) := \frac{1}{m} \sum_{k=1}^{m} y_k \log(\frac{x_k}{\sum_{l=1}^{l} x_l})$. At inference time, suppose the trained parameters is given by $W$. On a new instance $I$, the CG cut will be selected as $\arg\max_{i \in [m]} H_W(I, o_i)$, and ties are broken in an uniformly random manner.

Table 3: Average tree size on 250 test instances of the GNN trained using either the B&C tree size or gap closed as a proxy vs. four simple benchmarks. **For all benchmarks except the GNN one, only the branch-and-cut tree size is reported, as it represents the final quantity of interest in this experiment.**

(a) Set cover

| Setting | B&C tree | gap closed |
|---|---|---|
| Optimal | – | 4.95 |
| Parallelism | – | 8.29 |
| Efficacy | – | 9.90 |
| Mix | – | 9.30 |
| Random | – | 9.71 |
| GNN | 8.27 | 8.65 |

(b) Facility location

| Setting | B&C tree | gap closed |
|---|---|---|
| Optimal | – | 86.31 |
| Parallelism | – | 144.09 |
| Efficacy | – | 123.63 |
| Mix | – | 133.72 |
| Random | – | 152.46 |
| GNN | 128.85 | 134.61 |

(c) Knapsack

| Setting | B&C tree | gap closed |
|---|---|---|
| Optimal | – | 425.44 |
| Parallelism | – | 576.55 |
| Efficacy | – | 582.83 |
| Mix | – | 572.55 |
| Random | – | 583.43 |
| GNN | 570.55 | 571.34 |

(d) Vertex cover

| Setting | B&C tree | gap closed |
|---|---|---|
| Optimal | – | 8.54 |
| Parallelism | – | 10.15 |
| Efficacy | – | 9.36 |
| Mix | – | 9.55 |
| Random | – | 10.10 |
| GNN | 9.70 | 9.71 |

