# OpenReview forum: "How hard is learning to cut? Trade-offs and sample complexity"
_ICLR.cc/2026/Conference — ICLR 2026 Poster_

### Official Review · Reviewer_hSqs · 2025-10-30

**Soundness:** 2
**Presentation:** 3
**Contribution:** 2
**Rating:** 4
**Confidence:** 3

**Summary:**

The paper provides the first sample-complexity lower bounds for the “learning-to-cut” framework in branch-and-cut MILP solvers. It proves that, for neural-network policies mapping an ILP instance to a Chvátal–Gomory cut, minimising either tree-size or gap-closed requires
Ω(VCdim(F)/ε) samples, i.e. as many samples as learning any generic real-valued target with squared loss. The bounds are shown tight up to log factors for ReLU nets. Complementary experiments on four NP-hard problems indicate that a GNN trained on the cheap “gap-closed” proxy achieves tree-size reductions close to the expensive oracle, corroborating the theoretical insight that the two scores are equally learnable.

**Strengths:**

1. Theoretically pioneering: First distribution-free lower bounds for learning-to-cut; closes the loop with recent upper bounds [Balcan et al. 2021, Cheng et al. 2024].
2. Generality: Results hold for any encoder–network family that satisfies minimal closure assumptions; not restricted to CG or tableau cuts.
3. Tightness: Lower bound Ω(WL log(W/L)/ε) matches the best-known upper bound Õ(WL/ε²) for ReLU nets, showing no exponential gap.
4. Empirical validation: Carefully designed GNN experiments demonstrate that gap-closed is a practical surrogate for tree-size, bridging theory–practice.
5. Clarity: Proof roadmap and supplementary document are rigorous and well written.

**Weaknesses:**

1. Restricted concept class: The lower bounds are proved only for single-cut selection, whereas modern solvers add batches (rounds) of cuts. It remains unclear whether the lower bound still holds when the learner outputs a *set* of cuts.
2. CG-only cutting regime: The lower bounds rely exclusively on Chvátal–Gomory (CG) cuts. Extending the results to more general cut families such as split cuts, GMI cuts, or knapsack cuts would significantly strengthen the theoretical claims.
3. Instance distribution: overly general setting: The lower bounds are worst-case over all possible distributions, without considering structured or parametric families (e.g., IPs generated from stochastic block models or Gaussian-distributed `A, b`). This may lead to overly pessimistic bounds.
4. No information-theoretic upper bound for gap-closed: While the authors conjecture that only cuts separating the fractional solution matter, they do not provide a distribution-dependent upper bound based on this insight. Thus, the sample complexity of gap-closed remains only partially characterized.
5. Limited empirical scope: Experiments are conducted only on small synthetic instances (≤ 50 variables). There is no validation on medium-scale realistic benchmarks such as the “easy” subset of MIPLIB 2017, leaving open whether the proxy quality degrades on real-world IPs.
6. Missing baselines: The empirical evaluation lacks comparison with recent learning-based cut selectors, such as LLM-guided methods (LIFT-FE, TransGPT-FE) or imitation learning with tree-size experts (Paulus et al., 2022), making it hard to assess the relative strength of the proposed approach.

**Questions:**

1. Do the lower bounds hold for batch cut selection?
   If the learner outputs a set of cuts instead of a single one, does the sample complexity lower bound still apply? Is a new theoretical framework needed?
2. Can the bounds be extended to general cut families?
   Is it possible to generalize the lower bounds beyond CG cuts to split cuts, GMI cuts, or other stronger cut classes?
3. Can tighter bounds be obtained under structured distributions?
   If instances are drawn from structured generative models (e.g., stochastic block-model IPs or Gaussian `A, b`), can we prove tighter sample complexity bounds?
4. Can we derive a distribution-dependent upper bound for gap-closed?
   Can the conjecture — that only cuts separating the fractional solution matter — be leveraged to derive a tighter, distribution-dependent upper bound on the sample complexity of learning with gap-closed?

---

> ### Author Response · Authors · 2025-11-20
>
> Thank you for your feedback. We have carefully addressed all of your comments and questions (in order) and would be delighted to engage in any further discussion if you find it helpful.
>
> $\textbf{Weaknesses:}$
>
> - 1. This is addressed in our response to Question 1 below.
>
> - 2. This is addressed in our response to Question 2 below.
>
> - 3. This is addressed in our response to Question 3 below.
>
> - 4. This is addressed in our response to Question 4 below.
>
> - 5. The paper is admittedly approaching the learning to cut question from the theory side. Indeed, our experiments are intended to align with our findings that show different cut scores do not have a theoretical influence in the sample complexity lower bounds. We agree with the reviewer that more computational experiments on the MIPLIB would help the generalization aspect of the computation and we plan them for future work.
>
> - 6. We agree with the reviewer. Nevertheless, our focus was not on the computational side but on the theoretical aspects, the sample complexity lower bound.
>
> $\textbf{Questions (Part I):}$
>
> - 1. Thanks for this subtle question. Our results are indeed stated for only one cut at the root.  Although we did not include an explicit statement, we can adapt the approach to a sequence of cuts (with the same lower bound), by stating how this separation power would evolve when adding consecutive cuts that are chosen via the concept class.
> The gist of our approach is to use transfer bounds from ``separation power'' of the cuts via the score proxy (measuring by the fat-shattering dimension) to sample complexity lower bounds.
> To see how we can adapt our proof to a sequence of cuts, we can use the same cut as for the same instances of the proof of the Transfer Lemma. This is possible as we are searching for lower bounds, not upper bounds: find a sequence of cuts that does the job (i.e., does the shattering part) suffices.
> In other words,  since there is a CG cut that leaves the instance intact in terms of admissible region (although one constraint is added), we can re-use the exact same cut in later rounds. On the other hand, since the other cut allows to increase the optimum, we can use it again to have an objective value that is at least as good, and the overall jump in objective value would be the same as one cut. Therefore, this approach allows to transfer our sample complexity lower bounds directly to a sequence of cuts.
> We did not include such a lower bound into a statement, because we believe it gains value only if it incorporates the number of cuts considered, and we believe this question to be a more challenging one and would deserve an additional study. In answer to the last part of your question, we do believe that an entirely new approach should be developed to design $\textit{strong}$ lower bounds for sequence of cuts (with respect to the number of cuts).
>
>
> - 2. Our results are indeed limited to CG cuts for now. In our analysis, we make use of the fine grained knowledge of how CG cuts change on specific instances. We suspect that similar lower bounds hold for other cuts, but our analysis should be changed, at least we do not see how to directly adapt our method to other cuts such as split cuts, GMI cuts, cross-polytope cuts, or cuts derived from cut-generating functions. It was natural to start from CG cuts, the most classical family, at the very core of integer programming development.
>
> - 3. Thanks for the thoughtful and stimulating  question. The question of distribution-dependent sample complexity is a fundamental and challenging one in learning theory. We believe that in the case of learning to cut, tighter bounds should exist for specific distributions.  For distribution-dependent lower bounds, our approach should be adapted, if not entirely changed: in that case, only the instances from a specific distribution should be considered, and there is no guarantee that we can perform shattering using constraints as we did.  A good starting point can be uniform distributions on Knapsack instances of a certain size. For tighter $\textit{upper bounds}$, please see the answer below.
>
>
> - 4. For upper bounds, since most results in sample complexity rely on \emph{uniform convergence}, the situation is slightly different. As you point out, it could be very well the case that the upper bounds become lower (while remaining informative). For example, we strongly suspect that Knapsack instances have a much milder sample complexity than the one derived from uniform convergence [Cheng et al., 2024, Sample complexity of algorithm selection using neural networks and its applications to branch-and-cut].

---

> ### Author Response · Authors · 2025-11-20
>
> $\textbf{Questions (Part II)}$
>
> - 4 (second question) Thanks for the thoughtful question. First, we would like to bring some clarification on the conjecture you refer to. From the point of view of the branch-and-cut tree score, there could be cuts that do not remove the fractional solution that lead to smaller trees, so for that case the situation remains subtle.
> From the point of view of the gap closed score, only cuts that cut off  the fractional solution can stricly improve the objective value. Therefore a pathway to improve the upper bounds is to perform an analysis on the new number of regions in the \emph{weight space} of the CG cut that lead to piecewise constant regions for that score.  This observation is based on the proof technique used in [NeurIPS 2021, Balcan et al., Sample complexity of tree search configuration: Cutting planes and beyond.] and [NeurIPS 2024, Cheng  et al., Sample complexity of algorithm selection using neural networks and its applications to branch-and-cut].
> Furthermore, it is also true that the \emph{cuts from the tableau} do remove the fractional solution, so this observation could also lead to improved upper bounds for cuts from the Tableau.

---

### Official Review · Reviewer_dRbm · 2025-11-01

**Soundness:** 3
**Presentation:** 3
**Contribution:** 3
**Rating:** 8
**Confidence:** 3

**Summary:**

This paper studies lower bounds on the sample complexity of learning cut policies for Branch and Cut solvers for integer linear programs.
A key step in the Branch and Cut framework is adding a cutting plane, which is an additional constraint that tighten the constraint set without eliminating any integral solutions.
There are many valid cutting planes to choose from, so traditional B&C implementations have a collection of hand-crafted heuristics designed to choose cuts that lead to fast solution times.
Recently a line of work has explored the idea of learning cutting plane selection policies from data: given a training collection of ILP instances sampled from some distribution, each instance is solved and the resulting B&C trees are used to train cut selection policies that optimize a utility metric attempting to capture the running time required to solve the problem after the cut is applied.
The two most commonly considered metrics are tree-size (the actual size of the resulting B&C tree, which is strongly predictive of the runtime), and the gap-closed metric, which measures the reduction in the optimality gap after introducing the cut.
Some prior work has provided upper bounds on the sample complexity of learning such cutting policies when the utility metric is the size of the resulting tree (but the authors claim that it can be generalized to the gap-reduction metric straight forwardly).

This paper has two main high-level contributions:
1. The authors derive sample complexity lower bounds for learning cut selection policies from data under fairly realistic assumptions. In particular, together with the upper bounds from prior work, they establish that the sample complexity is comparable for both the the gap-closed and tree-size metrics.
2. The authors empirically find that the gap-closed metric is a suitable proxy for the tree-size metric. This is useful because the gap-closed metric is much more computationally efficient to evaluate when generating training data.

Taken together, these results support the use of gap-closed for learning cut selection policies. It is not harder to learn than the tree-size metric from a sample complexity point of view, still provides a reasonable proxy for solution time, and is easier to calculate.

**Strengths:**

The problem of learning cut selection policies for the branch and cut framework is interesting and very practical, and this paper provides the first lower bounds on sample complexity for this problem.
The findings that the gap-reduction metric is a suitable approximation to tree-size also useful.
Overall I found the paper well written and feel like the main contributions were clearly communicated.

**Weaknesses:**

I would have liked for more of the key sample complexity lower bound argument to be sketched in the main body of the paper.
For example, in the discussion of Proposition 3.4, it is stated that the approach ignores $n \times m + m$ of the inputs, and I was curious to understand why.
There are also a number of minor typos throughout the paper, and a few significant ones in the experimental results (unless I have misunderstood something).

I think at the beginning of Section 2.2, the authors could include a little bit more detail about the notation. At first I had some trouble identifying the role played by the functions $f$ and $h$ in equation (2).

A few of the minor typos I noticed:
- Line 120: "Ou results" -> "Our results"
- Line 169: "that can derived at any iteration" -> "that can be..."?
- Line 181: Maybe "so this motivates the idea of learning such a score"?

**Questions:**

- On line 308, should it say "returns a vector in $[0,1]^m$" instead of $[0-1]^m$?
- On line 309, should it say that the function is "continuous at 1/2"?
- Table 2 seems to be missing some results for Facility Location compared to Table 3 in the appendix (i.e., the first column is all dashes instead of having an entry for GNN).
- In both Table 2 and Table 3, should the second column be "gap closed" and the third column be "B&C tree". The titles aren't consistent, but if I understand what is being reported, it seems like the dashes should always appear in the "gap closed" column.
- In Tables 2 and 3, since only GNN has a gap-closed variant, I wonder if it would make sense to have a single column and just include two versions of GNN: "GNN-Tree-Size" and "GNN-Gap-Closed".

---

> ### Author Response · Authors · 2025-11-20
>
> Thank you very much for your generous and encouraging assessment of our article and results. We have carefully addressed all of your comments and questions (in the same order), and would be delighted to engage in any further discussion if you find it helpful.
>
> $\textbf{Weaknesses}$
>
>
> - Regarding the proof sketch, we were willing to include it in the main body but decided to expand on other aspects and leave it in the appendix. Concerning  your question about the inputs being ignored: the ILP instances are represented by $n \times m + n + m$ (real) entries, however in our approach we are only using one constraint vector of the instance, which means the other entries are not being considered. It could turn out that the shattering dimension is greater, by handling correctly the remaining entries of the instance to $\gamma$-shatter the instances. This should allow to obtain  stronger lower bounds with respect to $n$ and $m$. However, we think that with respect to the rest of the parameters of the model (in the case of neural networks for example) our bounds are close to be tight.
> Also, thank you for reporting those typos. We address them below and in our updated version.
>
>
> - Thank you for pointing that Section 2.2 deserves more clarity. We reformulated this part by introducing some more detail in the notations in the updated version.
>
>
> -  Thanks for catching those typos. We took take of them in the updated version.
>
>
> $ \textbf{Questions}$
>
> - Agreed, thank you. ($\textit{in}$ is typically reserved for intervals in english, and $\textit{at}$ is used for continuity at a point).
>
> - Thanks for catching this. We indeed forgot to report one of the last numbers in the Table. The updated version now contains it.
>
> - Yes, the first column should always be B\&C tree size, and the second one should be gap closed, which we included in the updated version. There is also indeed a typo in the title of the columns. The dashes mean that the first column, i.e., the ``B\&C tree size'' is the same (and somehow irrelevant) for all other baselines apart from GNN, since the value does not change from the second column on.
>
> - Thank you for the suggestion. We agree that given our remark above, there should be a way to compress the display of the tables even more, but we believe that the current form remains, satisfactory, in order to insist that two separate scores are being used.

---

### Official Review · Reviewer_Fk2R · 2025-11-02

**Soundness:** 3
**Presentation:** 2
**Contribution:** 2
**Rating:** 4
**Confidence:** 3

**Summary:**

This paper provides the first sample complexity lower bounds for learning to select cutting planes in integer programming. The bounds apply to two performance scores: branch-and-cut tree size and gap closed. The authors show these lower bounds are nearly tight with known upper bounds for neural networks, indicating both scores have comparable learning difficulty. Empirical results demonstrate that gap closed serves as a practical proxy for tree size reduction when training graph neural networks on tableau Chvatal-Gomory cuts.

**Strengths:**

1. First sample complexity lower bounds for learning-to-cut, establishing theoretical foundations
2. Lower bounds hold for wide function classes and are nearly tight with upper bounds up to logarithmic factors
3. Theoretical equivalence between scores provides formal justification for using gap closed as proxy
4. Proof construction for shattering ILP instances is non-trivial and well-executed
5. Experiments directly test the core thesis about proxy effectiveness

**Weaknesses:**

1. Gap between $\Omega(1/\epsilon)$ lower bound and $\Omega(1/\epsilon^2)$ upper bound for tableau case
2. Assumptions 1 and 2 restrict generality of theoretical claims
3. Empirical validation uses small-scale problems in controlled environment and restricted solver configuration
4. Performance varies: on Facility Location, Efficacy heuristic (123.63) outperforms GNN (134.61)
5. Experimental setup disables key solver components (presolve, heuristics, default cuts)
6. Theoretical analysis confined to Chvatal-Gomory cuts from simplex tableau
7. Learned policy selects single cut at root node, not dynamic cut selection
8. Worst-case bounds do not leverage structure in real-world MILP distributions
9. Computational cost of obtaining training samples not addressed theoretically

**Questions:**

1. Can the $\epsilon$-dependence gap be closed? Is the true sample complexity $\Theta(1/\epsilon)$ or $\Theta(1/\epsilon^2)$?
2. How would sample complexity scale for learning policies that select multiple rounds of cuts?
3. Would results transfer to production solver configurations with presolve and heuristics enabled?
4. Do similar lower bounds hold for other cut families beyond Chvatal-Gomory cuts?

---

> ### Author Response · Authors · 2025-11-20
>
> Thank you for your feedback. We have carefully addressed all of your comments and questions (in those three messages) and would be delighted to engage in any further discussion if you find it helpful.
>
> $\textbf{Weaknesses (Part I):}$
>
> -  1. We are willing to engage in a more detailed exchange, as  we fail to see how the mentioned gap in the table constitutes a weakness in itself. First, as detailed in our first answer to your question in the second message below, the gap between $\frac{1}{\epsilon}$ and $\frac{1}{\epsilon^2}$ is unavoidable in a supervised learning setup with neural networks. Second, with respect to the number of parameters of the neural networks, the gap between the two upper and lower bounds is almost tight in the regime where the number of parameters of  the first layer is small compared to the total number of parameters.
>
>
> -  2. Assumptions 1 and 2 serve as formal mathematical guardrails. We would like to point out that they are fairly general: for example, they are valid for machine learning models such as neural networks, graph neural networks, etc.
> They remain valid for more recent models such as transformers. Furthermore, it is easy to see that a statement for lower bounds for any machine learning model (without any further assumption) is  impossible: imagine a machine learning model that simply ignores the constraint of the integer program, and another that verifies our assumption, then the former  one has zero sample complexity (any model of the class returning always the same output) and the latter verifies our lower bounds.
> We believe that Assumption 2 may generate some confusion, and we would like to justify its usage here. Consider the extreme example of a machine learning method that for some reason ignores a certain rows of the integer program instances. Then, our approach for that model would directly fail as we need the model to shatter a certain row (the one of the redundant constraint) with a certain shattering power.  A less extreme example would be the case of a model that contains more shattering power over certain rows, and very little for certain ones, for example the one corresponding to the redundant constraint we use for shattering purposes. Our statement could be adapted to that case, but would make use of the weak shattering power of the model along this constraint. For our result to be significant and crisper, in the sense that it represents the overall model's shattering power, we made this assumption of ``non favored'' constraint, which felt quite natural to us.
> In other words, we think that our results can be stated without Assumption 2 by replacing it with another statement, but we believe that it would become less crisp than under its current form. Also, since many machine learning models verify Assumption 2, we decided to keep it under this form.
>
>
> -  3. We agree that our experiments are limited to thousand of instances, for various types of problems (an extended presentation of the experiments  is included in the appendix). To the best of our knowledge, a  large scale (e.g., for hundreds of thousand or millions of instances) benchmark of data driven techniques for learning to cut does not exist in the literature. We think this would constitute a very valuable complement to our results.
>
> - 4. The reviewer's observation is clearly correct. However, the goal of our experiments is to show that gap closed and number of nodes are similar in terms of learning power as score function for cut selection. The goal is motivated by the sample complexity analysis that deems them equivalent. So, the interpretation of the results is that the GNN is able to learn with both scores and this conclusion is reached by comparing the GNN with classical heuristics for cut selection. In other words, we are not proposing a new selection method, we do not consider the fact that the GNN does not dominate the baselines as a weakness.
>
>
> - 5. Disabling those solver components is standard for this type of analysis. The reasons are: 1) presolve would ``kill'' most of the instances of that size, making the experiments meaningless, 2) heuristics would introduce a form of randomness in the process because finding or not a primal solution depends on performance variability components like the optimal basis of the simplex tableau, and 3) other cuts would collide with the single cuts selection we are analyzing. Nevertheless, we are convinced that if those components are used $\textit{a priori}$ just for strengthening the initial formulation, the cut analysis would be consistent.

---

> ### Author Response · Authors · 2025-11-20
>
> $\textbf{Weaknesses (Part II)}$
>
> - 6. Our results are indeed limited to CG cuts for now. In our analysis, we make use of the fine grained knowledge of how CG cuts change on specific instances. We suspect that similar lower bounds hold for other cuts, but our analysis should be changed, at least we do not see how to directly adapt our method to other cuts such as cross-polytope cuts, or cuts derived from cut-generating functions. We believe it was natural to start from CG cuts, the most classical family, at the very core of integer programming.
>
>
> -  7. Indeed, our analysis restricts to a single cut at the root node, as we believe this case is already a challenging question, for which no prior lower bound existed. We would like to add that our approach can yield (a non tight) lower bound in the case of dynamic cut selection.
> Although we did not include an explicit statement, we can adapt the approach to a sequence of cuts, by stating how this separation power would evolve when adding consecutive cuts that are chosen via the concept class. Our approach is then the following. We use the same cuts as for the instances of the proof of the Transfer Lemma: Since there is a CG cut that leaves the instance intact in terms of admissible region (although one constraint is added), we can re-use the exact same cut in later rounds. On the other hand, since the other cut allows to increase the optimum, we can use it again to have an objective value that is at least as good, and the overall jump in objective value would be the same as one cut.
> Therefore, this approach allows to transfer our sample complexity lower bounds directly to a sequence of cuts. However, it would have to be refined to include a dependence on the number of cuts, which we believe to be a more challenging problem.
>
>
> - 8. We entirely agree with this point. In our work we do not restrict attention to any particular structure of the MILPs that might yield weaker lower bounds; instead, our results are distribution-independent and therefore necessarily more general. Exploring more specialized families of instances - such as Knapsack -- is a natural direction for future work. We suspect that, under such structural restrictions, the lower bounds could in fact be significantly smaller. Establishing this formally, however, would require revisiting and adapting several parts of our proof technique, since our current arguments rely crucially on hardness constructions that are not tailored to Knapsack structure.
> In addition, the corresponding upper bounds for specific instances would also merit investigation. Understanding whether specialized algorithms (or hierarchies) can exploit this structure to outperform the worst-case, distribution-independent guarantees, is a very interesting follow-up investigation.
>
>
> - 9. We completely agree that obtaining training samples for data-driven techniques in optimization can be computationally costly, and that we do not address this aspect in our article. The premise of our study is that for instances that are of reasonable size (hundreds of constraints / variables), the cost of data collection remains manageable and we wish to provide solid theoretical basis for a more systematic usage of data-driven decision making in optimization.

---

> ### Author Response · Authors · 2025-11-20
>
> $\textbf{Questions}$
>
> - 1. The best known results with respect to the dependency in $\epsilon$ is $\Omega(\frac{1}{\epsilon})$ for lower bounds vs. $\mathcal{O}(\frac{1}{\epsilon^2})$ for the corresponding upper bounds. This holds in the fundamental supervised  setup, with neural networks (cf. [Anthony \& Bartlett, 2009, Neural Network Learning: Theoretical Foundations, in particular discussions in Section 19.5]). In that case,  the gap is known to be unavoidable in the case where there is no assumption made on the distribution (uniform sample complexity).  We suspect this is also the case in our setup, but we would leave this for future work.
>
> - 2. Thanks for this subtle question. Our results are indeed stated for only one cut at the root.  Although we did not include an explicit statement, we can adapt the approach to a sequence of cuts (with the same lower bound), by stating how this separation power would evolve when adding consecutive cuts that are chosen via the concept class.
> The gist of our approach is to use transfer bounds from ``separation power'' of the cuts via the score proxy (measuring by the fat-shattering dimension) to sample complexity lower bounds.
> To see how we can adapt our proof to a sequence of cuts, we can use the same cut as for the same instances of the proof of the Transfer Lemma. This is possible as we are searching for lower bounds, not upper bounds: find a sequence of cuts that does the job (i.e., does the shattering part) suffices. In other words,  since there is a CG cut that leaves the instance intact in terms of admissible region (although one constraint is added), we can re-use the exact same cut in later rounds. On the other hand, since the other cut allows to increase the optimum, we can use it again to have an objective value that is at least as good, and the overall jump in objective value would be the same as one cut. Therefore, this approach allows to transfer our sample complexity lower bounds directly to a sequence of cuts.
> We did not include such a lower bound into a statement, because we believe it gains value only if it incorporates the number of cuts considered, and we believe this question to be a more challenging one that would deserve an additional study.
>
> - 3. Our experiments included additional runs with Gurobi’s default presolve and primal heuristics enabled, and we observed that the results were very similar to those reported in the paper. That said, transferring these findings to full-scale industrial solver configurations is less straightforward. Production environments often involve highly customized parameter settings, problem-specific tuning, and proprietary heuristics. Evaluating the robustness of our cuts under such heterogeneous configurations would require a larger and more systematic study that explores a broad range of solver settings. This remains an important direction for future work.
>
>
> - 4. In our analysis, we make use of the fine grained knowledge of how CG cuts change on specific instances. We suspect that similar lower bounds hold for other cuts, but our analysis should be changed, at least we do not see how to directly adapt our method to other cuts such as cross-polytope cuts, or cuts derived from cut-generating functions. As stated, starting from CG cuts, the most classical family in integer programming, looked like the natural choice.

---

### Official Review · Reviewer_BtH5 · 2025-11-04

**Soundness:** 3
**Presentation:** 3
**Contribution:** 3
**Rating:** 6
**Confidence:** 3

**Summary:**

The paper proves sample complexity bounds in the setting of learning to cut for solving integer programs. The bounds are for two different scores, branch and cut tree size and gap closed. The theoretical results show that both scores are similarly difficult to learn. Additionally, the paper provides

**Strengths:**

- The paper is well written: it illustrates its core ideas through intuitive examples and explanations without sacrificing technical details.
- The bounds derived don't seem to be vacuous/uninformative. The proofs are also relying on (as far as I can tell) novel constructions that are specific to the problem.
- To the best of my knowledge, these are the best known nontrivial bounds for this learning problem.
- The paper also provides an experimental comparison of the learned cuts of a GNN compared to well known heuristics and shows that the GNN can learn to generate cuts that are on par or better compared to those heuristics.

**Weaknesses:**

- In the proof of 3.2 (Around line 600), you set $a = \epsilon$ and use the Transfer Lemma.
Doesn't that mean you take $$\text{fat} _{\mathcal{F} _{s,\sigma'}} (\frac{\epsilon}{\epsilon}) \geq \text{VCdim}(\mathcal{F}[n]) ?$$
However the transfer lemma holds for $\gamma \in (0,1/2)$ so I'm not sure you want to do it this way. Maybe set $a=c\epsilon$ for c at least 2?
- How necessary is assumption 2 for your result? Are there any architectures that don't qualify? That's perhaps one part I'm a little unclear about and I woud like to see some more comments on.
- There are a bunch of typos in the document so more careful proofreading is required. For example see lines: 120, 224, 459-460,
- I am not sure how the experimental section ties into the overall contribution here. I guess it's good to see that GNNs can learn a practically viable heuristic by training on gap closed but it is a bit unclear how that connects to the theoretical findings of the paper.

Overall this is a good paper that I lean towards accepting. There are some (I think minor) issues with the proof and I'm a bit unclear about assumption 2 so I start with a tentative score that I will reconsider after the rebuttal.

**Questions:**

see above

---

> ### Author Response · Authors · 2025-11-20
>
> Thank you for your positive assessment of our article and results. We have addressed all of the points raised in the same order, and would be glad to engage in further discussion if that would be helpful.
>
> $\newline$
>
> $\textbf{Weaknesses and questions}$
>
>
> - Thanks for catching this typo. Around line 600, was meant   $\alpha = 4\epsilon$ (instead of $\alpha = \epsilon$) with $\epsilon < \frac{1}{16}$ so that $ \frac{\epsilon}{\alpha} (=\frac{1}{4}) \in (0, \frac{1}{2})$, which allows us to use the Transfer Lemma without any additional change. Also, we would like to add that in Line 596, the inequality should have on its right hand side a denominator of $ 16 \alpha$ instead of $\alpha$, according to the statement of Theorem 2.3, leading to a smaller constant in our final bound.
> We have corrected this typo in the updated version.
>
>
>
> - Thank you for this interesting question. First consider the extreme example of a machine learning method that for some reason ignores a certain rows of the integer program instances. Then, our approach for that model would directly fail as we need the model to shatter a certain row (the one of the redundant constraint) with a certain shattering power.  A less extreme example would be the case of a model that contains more shattering power over certain rows, and very little for certain ones. For the result to be somehow significant and to be representative of the model's shattering power, we made this assumption of ``non favored'' constraint, which felt quite natural to us.  However, we think that our results can be stated without Assumption 2 via other statements, though we believe that the overall statement could be more confusing or less crisp than under its current form. Also, since many machine learning models verify Assumption 2, we decided to keep it under this form.
>
>
>
> - Thank you for reporting those typos. We corrected the ones you indicated and did another proofreading of the document.
>
>
>
> - One of the driving motivations of this study is to compare the pertinence and relative difficulty of using one score or the other in integer programming for learning to cut (gap closed and branch-and-cut tree size). In our experiments, we notice little difference in the use of one score or the other for training with respect to the true tree size improvement. This observation aligns well with the bounds we obtain that also seem to be independent of the score used: for example, the known upper bounds are valid for both scores, and our lower bounds hold as well (in the case of neural networks, those bounds are almost tight). Therefore, we would like to argue that the  message of the experiments is seen as complementary to our theoretical results, as it provides some clue on the pertinence and difficulty of using one score or the other for learning to cut.

---

### Meta-Review · Area_Chair_N1LH · 2026-01-07

**Summary:**

The paper establishes sample complexity lower bounds for the problem of cutting plane selection, arguing that the theoretically comparable learnability of two core functions, i.e. branch-and-cut tree size and gap closed, justifies using the former as a computational proxy for the latter. The paper supports this through having similar lower bounds and through empirical experiments using graph neural networks on set covering and facility location instances to demonstrate the efficacy of the gap-closed metric. The reviews are mixed in this case, so I can understand if the decision needs to be changed, but I eventually believe this is an interesting observation and will likely be relevant for the literature. Nevertheless, I strongly recommend the authors to revise the paper and include all the recommendations of the reviewers in the camera-ready and conduct the additional experiments (baselines and benchmarks) requested, since this will strengthen the claims.

**Reviewer Concerns:**

The paper originally received various questions and weaknesses on the writing, with various reviewers mentioning proofreading. In addition, various questions focus on the real world validity of the claims and how those are connected with practical insights and utility. Some additional concerns are around the limited experimental evaluation and lack of baselines (Reviewer hSqs). The rebuttal does not offer any new experiments, while it reiterates that the theoretical contributions are the important ones. I do not believe that this would have been a convincing argument for Reviewer hSqs. I do find this is a valid argument and hopefully the camera-ready version of the paper can have stronger experiments, since those are small scale experiments. Another concern was raised on extending the results on more general cut families. The rebuttal does not offer any additional results there.

**Reviewer Scores:**

The two positive reviews, i.e. from Reviewer dRbm and from Reviewer BtH5, have various questions that were addressed during the rebuttal. Having said that, I am not sure whether the scores would have changed for those two reviews, since they are already quite positive.

On the other hand, for Reviewer hSqs, the authors have provided answers, but I am not sure whether those would have convinced the reviewer. Overall, this is among the papers, where a complete period of discussion would have helped, but I do think this is an interesting theoretical paper and I hope the authors can do the requested revisions in the camera-ready version.

---

### Decision · Program_Chairs · 2026-01-26

Accept (Poster)